

# Morphological and molecular data confirm the transfer of homostylous species in the typically distylous genus *Galianthe* (Rubiaceae), and the description of the new species *Galianthe vasquezii* from Peru and Colombia

Javier Elias Florentín[1], Andrea Alejandra Cabaña Fader[1], Roberto Manuel Salas[1], Steven Janssens[2], Steven Dessein[2] and Elsa Leonor Cabral[1]

[1] Herbarium CTES, Instituto de Botánica del Nordeste, Corrientes, Argentina
[2] Plant systematic, Botanic Garden Meise, Meise, Belgium

Corresponding author
Javier Elias Florentín,
florentinjaviere@gmail.com

## ABSTRACT

*Galianthe* (Rubiaceae) is a neotropical genus comprising 50 species divided into two subgenera, *Galianthe* subgen. *Galianthe,* with 39 species and *Galianthe* subgen. *Ebelia*, with 11 species. The diagnostic features of the genus are: usually erect habit with xylopodium, distylous flowers arranged in lax thyrsoid inflorescences, bifid stigmas, 2-carpellate and longitudinally dehiscent fruits, with dehiscent valves or indehiscent mericarps, plump seeds or complanate with a wing-like strophiole, and pollen with double reticulum, rarely with a simple reticulum. This study focused on two species that were originally described under *Diodia* due to the occurrence of fruits indehiscent mericarps: *Diodia palustris* and *D. spicata*. In the present study, classical taxonomy is combined with molecular analyses. As a result, we propose that both *Diodia* species belong to *Galianthe* subgen. *Ebelia*. The molecular position within *Galianthe*, based on ITS and ETS sequences, has been supported by the following morphological characters: thyrsoid, spiciform or cymoidal inflorescences, bifid stigmas, pollen grains with a double reticulum, and indehiscent mericarps. However, both species, unlike the remainder of the genus *Galianthe*, have homostylous flowers, so the presence of this type of flower significantly modifies the generic concept. In this framework, a third homostylous species, *Galianthe vasquezii*, from the Andean region is also described. Until now, this species remained cryptic under specimens of *Galianthe palustris* It differs however from the latter by having longer calyx lobes, the presence of dispersed trichomes inside the corolla lobes (vs. glabrous), fruits that are acropetally dehiscent (vs. basipetally dehiscent), and its Andean geographical distribution (vs. Paranaense). Additionally, a lectotype has been chosen for *Diodia palustris*, *Borreria pterophora* has been placed under synonym of *Galianthe palustris*, and *Galianthe boliviana* is reported for the first time from Peru. A key of all *Galianthe* species with indehiscent mericarps is also provided.

## INTRODUCTION

*Galianthe* Griseb. is a neotropical genus belonging to tribe Spermacoceae (*Groeninckx et al., 2009*). The genus was revised by *Cabral (2002)* and divided into two subgenera (*Cabral & Bacigalupo, 1997*): *Galianthe* subgen. *Galianthe*, from South America with 39 species, and *Galianthe* subgen. *Ebelia* (Rchb.) E.L. Cabral & Bacigalupo, with 11 Mesoamerican and South American species. Historically, *Galianthe* was associated with *Diodia* L., which has been described based on only *D. virginiana* L. The type species has a palustrine habit, pauciflorous axillary inflorescences, filiform corolla tube, bifid style with two long filiform stigmatic lobes, and indehiscent fruits. Later, others authors (i.e., *Swartz, 1788*; *Walter, 1788*; *Candolle, 1830*; *Small, 1913*) added other species into this genus with diverse kinds of habits and inflorescences, different floral morphology (e.g., distyly or homostyly, infundibuliform or campanulate corollas, bifid or bilobate stigma), and 2-carpellate schizocarpic fruits, being currently comprised by ca. 180 names (called *Diodia s. lat.*). Later, *Bacigalupo & Cabral (1999)* revised the genus *Diodia* and maintained only five species that were morphologically similar to the type species *D. virginiana* L. (description as above, and constituting *Diodia s. str.*). Species that did not match with these diagnostic features, were transferred to other genera as follows: eight species to *Borreria* subgen. *Dasycephala* (DC.) Bacigalupo & E.L. Cabral (*Bacigalupo & Cabral, 1996*), 12 species to *Hexasepalum* Bartl. ex DC. (*Kirkbride, 2014*; *Kirkbride & Delprete, 2015*; *Cabaña Fader et al., 2016*), and four species to *Galianthe* subgen. *Ebelia* (*Cabral & Bacigalupo, 1997*). The four *Galianthe* species are distylous, *Diodia bogotensis* (Kunth) Cham. & Schltdl. [=*Galianthe bogotensis* (Kunth) E.L. Cabral & Bacigalupo]; *D. brasiliensis* Spreng. [=*G. brasiliensis* (Spreng.) E.L. Cabral & Bacigalupo]; *D. cymosa* Cham. [=*G. cymosa* (Cham.) E.L. Cabral & Bacigalupo], and *D. hispidula* A. Rich. ex DC. [=*G. hispidula* (A. Rich. ex DC.) E.L. Cabral & Bacigalupo]. The remaining species with an uncertain position (ca. 150 names) are currently under revision (A Cabaña Fader, pers. comm., 2017). In this sense, (*Bacigalupo & Cabral, 1996*; *Bacigalupo & Cabral, 1998*) transferred these species to *Borreria* subgen. *Dasycephala* because of their homostylous flowers and indehiscent mericarps, while *Delprete, Smith & Klein (2005)* and *Delprete (2007)*, alluding to a broad concept, transferred the two species to *Spermacoce* mainly based on fruit characters. *Dessein (2003)* informally proposed to consider *Diodia spicata* as part of *Galianthe* based on molecular data (ITS intron), palynological data (double reticulum), and fruit morphology. The aim of this work is to confirm the taxonomic position of *D. palustris* and *D. spicata* based on morphological and molecular data, and perform their formal combination in *Galianthe*. In addition, a third homostylous species (*Galianthe vasquezii* R.M Salas & J. Florentín) is described and illustrated based on specimens from Colombia and Peru (previously identified as *D. palustris*). Additionally, a lectotype has been chosen for *Diodia palustris* whereas *Borreria pterophora* has been placed under synonymy of *Galianthe palustris*. Moreover, *Galianthe boliviana* E.L. Cabral is for the first time recorded in Peru. Finally, we provided a distribution map for the species investigated in this study, as well as a dichotomous key for all taxa with indehiscent mericarps.

## MATERIALS AND METHODS

### Morphological study

This study is based on classical taxonomy techniques. Collections deposited at the BA, BHCB, CEPEC, CTES, ESA, FUEL, FPS, FURB, HAS, HOXA, HUT, IAC, IAN, IFFSC, IPA, K, LIL, MBM, MO, NY, P, PR, SI, SP, UB, UFRN, USB, US, USM and UEC herbaria were analysed. Furthermore, the databases of the 'Catálogo de plantas e fungos do Brazil' and 'Missouri Botanical Garden' were examined. In order to carry out scanning electron microscope (SEM) analyses, flowers were dehydrated using a graded series of ethanol solutions and afterwards critically point dried and sputter-coated with gold-palladium. SEM images were obtained with a JEOL 5800 LV scanning electron microscope. Pollen grains were acetolyzed according to *Erdtman (1966)* and mounted in glycerine jelly for analysis by light microscopy (LM). Conventional parameters (P = polar axis, E = equatorial axis) of at least 20 grains were measured under LM, and the exine was analyzed using SEM. Pollen terminology follows *Punt et al. (2007)*. Species distribution maps were generated from distribution data that was present on the herbarium labels for each specimen and subsequently georeferenced using Google Earth (www.google.earth.com.ar) and *Hijmans (2013)*.

The electronic version of this article in Portable Document Format (PDF) will represent a published work according to the International Code of Nomenclature for algae, fungi, and plants (ICN), and hence the new names contained in the electronic version are effectively published under that Code from the electronic edition alone. In addition, new names contained in this work which have been issued with identifiers by IPNI will eventually be made available to the Global Names Index. The IPNI LSIDs can be resolved and the associated information viewed through any standard web browser by appending the LSID contained in this publication to the prefix "http://ipni.org/". The online version of this work is archived and available from the following digital repositories: PeerJ, PubMed Central, and CLOCKSS.

### Molecular study

In total, 45 species (47 accessions) were included to infer the phylogenetic relationship of *Diodia palustris* and *D. spicata*. The ingroup contains species from the *Borreria*, *Carajasia* R.M. Salas, E.L. Cabral & Dessein, *Crusea* Cham. & Schltdl., *Diodia, Emmeorhiza* Pohl ex Endl., *Ernodea* Sw., *Galianthe, Hexasepalum, Mitracarpus* Zucc., *Psyllocarpus* Mart. & Zucc., *Richardia* L., *Schwendenera* K. Schum., *Spermacoce*, and *Staelia* Cham. & Schldtl. genera, and *Bouvardia ternifolia* (Cav.) Schltdl. as the outgroup. Leaf samples of these studies were obtained from silica gel-dried material or herbarium materials. Forty-three species (44 accessions) were previously used by *Salas et al. (2015)*. Four accessions belonging to *D. palustris* has been added. All studied species with geographical information, collector, herbarium and GenBank accession numbers are provided in the Appendix.

### Molecular protocols

Total genomic DNA was isolated from silica-dried leaf material using a modified CTAB protocol (*Doyle & Doyle, 1987*). Nuclear ribosomal ETS and ITS fragments were amplified following *Baldwin & Markos (1998)* and *Negrón-Ortiz & Watson (2002)*, and

*White et al. (1990)*, respectively. PCR reactions for both gene markers investigated in this study consisted of 2 min initial denaturation at 94 °C and 30 cycles of 30 s denaturation at 94 °C, 30 s primer annealing at primer specific temperature and 1 min extension at 72 °C. Primer annealing for ETS and ITS were at 47 °C and 48 °C respectively. Amplification reactions were carried out on a GeneAmp PCR system 9700 (Applied Biosystems, Foster City, CA, USA). Purified amplification products were sent to Macrogen, Inc. (Seoul, South Korea) for sequencing. Sequences obtained in this study were deposited at GenBank [*Diodia palustris, Verdi et al. 1905*, ETS (MF166824), ITS (MF166826); *Miguel et al. 19*, ETS (MF166825), ITS (MF166827).

## Phylogenetic analyses

Contiguous sequences were assembled using Geneious v7.0.6 (Biomatters, Auckland, New Zealand). Automatic alignments were carried out with MAFFT (*Katoh et al., 2002*) Subsequent manual finetuning of the aligned dataset was done in Geneious v7.0.6. Congruency between the different datasets was inferred using different methods. First, a series of incongruence length difference tests (*ILD*; *Farris et al., 1995*) were carried out with PAUP* v.4. 0b10 (*Swofford, 2003*) using the following parameters: simple taxon addition, TBR branch swapping and heuristic searches of 1,000 repartitions of the data. Despite the well-known sensitivity of the ILD test (*Barker & Lutzoni, 2002*), the results of this test were compared in light of the resolution and support values of the obtained nuclear and nuclear ribosomal topologies. As a result, possible conflict between data matrices was visually inspected, searching for conflicting relationships within each topology that are strongly supported (hard vs. soft incongruence; *Johnson & Soltis, 1998*. Model selection for the Bayesian inference analysis was conducted with ModelTest 3.06 (*Posada & Crandall, 1998*) under the Akaike Information Criterium (AIC). The GTR+G model was selected for both ITS and ETS. Bayesian analyses of the concatenated dataset were carried out with MrBayes 3.1 (*Huelsenbeck & Ronquist, 2001*; *Ronquist & Huelsenbeck, 2003*). Four chains (one cold, three heated), initiated from a random starting tree were run simultaneously for 10 million generations. Every 1,000 generations, a tree was sampled from the chain for a total of 10,000 trees. Due to the burn-in, 50% of the sample points were discarded. Convergence of the chains was examined with TRACER 1.4 (*Rambaut & Drummond, 2007*). This resulted in an effective sampling size (ESS) parameter exceeding 100, which assumes a sufficient sampling and acceptable mixing.

## RESULTS

### Phylogenetic results

The ingroup contains 14 genera represented by 45 species of the *Spermacoce* clade. Of these, *Diodia spicata* and *D. palustris* are analysed for the first time in this context. ITS and ETS datasets were analysed both separately and combined. Because topology of each gene marker is very similar, we only present the results of the combined analysis (Fig. 1). Current results indicate that most clades coincide with most currently accepted genera (e.g., *Crusea, Emmeorhiza, Ernodea, Diodia s.s.* (sensu *Bacigalupo & Cabral, 1999*), *Mitracarpus, Psyllocarpus, Richardia* and *Staelia*). *Spermacoce, Borreria* and *Hexasepalum*

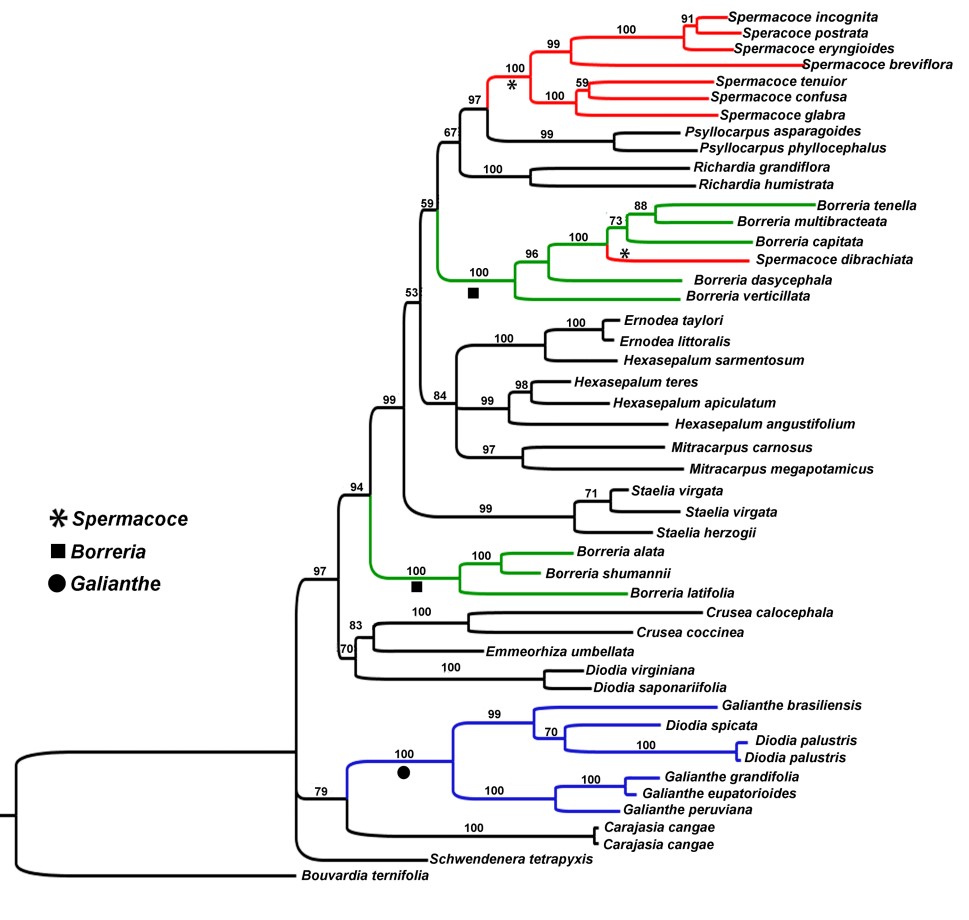

**Figure 1** **Bayesian tree.** Showing the relationship of *Galianthe* with the remaining genera of *Spermacoce* clade.

however, appear unresolved or as polyphyletic genera being present in several different parts of the tree. In regard of the species studied, we found that *G. spicata* and *G. palustris* fall intermingled among the *Galianthe* species. The *Galianthe* clade, including both former *Diodia* species, is strongly supported (Bayesian Posterior Probability (BBP): 100). The genus is divided into two strongly supported subclades, one subclade consists of *G. grandifolia* E.L. Cabral, *G. eupatorioides* (Cham. & Schltdl.) E.L. Cabral, and *G. peruviana* (Pers.) E.L. Cabral (BPP: 100), all from *G.* subgen. *Galianthe*. The other subclade (BPP: 99) comprises *G. spicata*, *G. palustris*, and *G. brasiliensis* (type species of *Galianthe* subgen. *Ebelia*). The clade of *Galianthe* and *Carajasia* is only moderately supported (BBP: 79). The genera *Galianthe* and *Carajasia* share the occurrence of pollen grains with a double reticulum, mostly associated with the distyly and bifid stigma. The *Galianthe-Carajasia* clade forms an unsupported trichotomy with *Schwendenera* (also distylous) and the remaining genera of the *Spermacoce* clade (all homostylous species never associated to double reticulum pollen grains). Interestingly, all clades that coincide with generic concepts are strongly supported (e.g., *Psyllocarpus* (BBP:99), *Spermacoce s.s.* (BBP:100), *Richardia* (BBP:100),

*Borreria s.s.* (BBP:100), *Mitracarpus* (BBP:97), *Hexasepalum s.s.*, *Staelia* (BBP:99), *Diodia s.s.* (BBP:100), *Borreria latifolia* group (BBP:100), and *Crusea* (BBP:100)). The species assigned to *Borreria* (sensu *Bacigalupo & Cabral, 1996*) are divided into two clades that are intermingled with other morphologically well-defined genera. One of these clades, further referred to as the *Borreria latifolia* group, comprises *Borreria alata*, *B. schumannii*, and *B. latifolia* (BBP:100). The other clade comprises five *Borreria* species from North and South America (*B. capitata*, *B. multibracteata*, *B. tenella*, *B. dasycephala*, and *B. verticillata*), as well as the African *Spermacoce dibrachiata* (BBP: 100). *Spermacoce* is divided into two unrelated branches, of which one clade comprises the type species *S. tenuior,* other American species with similar flower morphology (*S. eryngioides*, *S. prostrata*, *S. incognita*, *S. confusa*, and *S. glabra*, all with stamens and style included), and the Australian *S. breviflora* (support 100). As mentioned above, the other species of *Spermacoce* (*S. dibrachiata*) falls among the species of *Borreria*. *Hexasepalum* species are also divided into two clades, one of them is well supported (BBP: 99) and contain *H. angustifolium* Bart. ex DC. (type species), *H. apiculatum* and *H. teres*. The other, only represented by *H. sarmentosum* appears as sister species of the *Ernodea* (BBP: 100). The genus *Ernodea*, represented by *E. taylori* and *E. littoralis*, constitutes a strongly supported clade (BBP: 100). The results explained above allow us to support the following taxonomic changes.

## TAXONOMIC TREATMENT

### Description of the new species

***Galianthe vasquezii*** R. M. Salas & J. Florentín, ***sp. nov***. TYPE. PERU: Pasco, Oxapampa, Parque Nacional Yanachaga-Chemillen, Quebrada Yanachaga, 2,250 m, 10°24′S, 75°28′W, 14 Jun 2003, *R. Vásquez M. 28284* (holotype: HOXA!; isotypes MO!, HUT, USM).

#### Description

Herb decumbent or prostrate, stems quadrangular, angle strongly alate, with scabridous papillae, more densely disposed near nodes. Leaves sessile or pseudopetiolate, pseudopetiole up to 4 mm long, blades elliptic or obovate, apex acute, base attenuate, 12–32 × 5–17 mm, plicate-nervose, adaxially glabrous or puberulous, abaxially scabridous on nerves, margin scabridous, with 3–5 secondary nerves; stipular sheath 3.2–5.6 mm long, with 7–9 linear fimbriae, glabrous, fimbriae 3.5–6.8 mm long. Inflorescences thyrsoid, partial inflorescences subglomeriform, multiflorous. Flower pedicellate; pedicel 1–2 mm long; calyx (3-) 4-lobed, hypanthium 1.1–1.3 mm long, glabrous or glabrescent, lobes narrowly triangular, 1–1.4 mm long, glabrous, apex acute; corolla infundibuliform, 3-lobed, 1.75–2.1 mm long, white; lobes ovate, internally with hairs scattered at base, tube internally with some dispersed hairs near its base and externally glabrous, straight; stamens subincluded, anther 0.4–0.6 mm long, oblong, filament fixed immediately below interlobular sinuses; pollen grains 7–8 zonocolpate, oblate-spheroidal to prolate spheroidal, small, $P = 31$ μm, $E = 29$ μm, colpi long, endoaperture an endocingulum, exine semitectate, reticulate, muri nanospinose, 0,18–0,3 μm long; style bifid, 1.5–1.8 mm long, stigmatic branches ca. 0.2 mm long, with conspicuous papillae. Fruit a capsule, cordate or deltoid in outline, 1.8–2 × 1.6–1.9 mm, glabrous, with two indehiscent mericarps which split from the base upwards, each valve

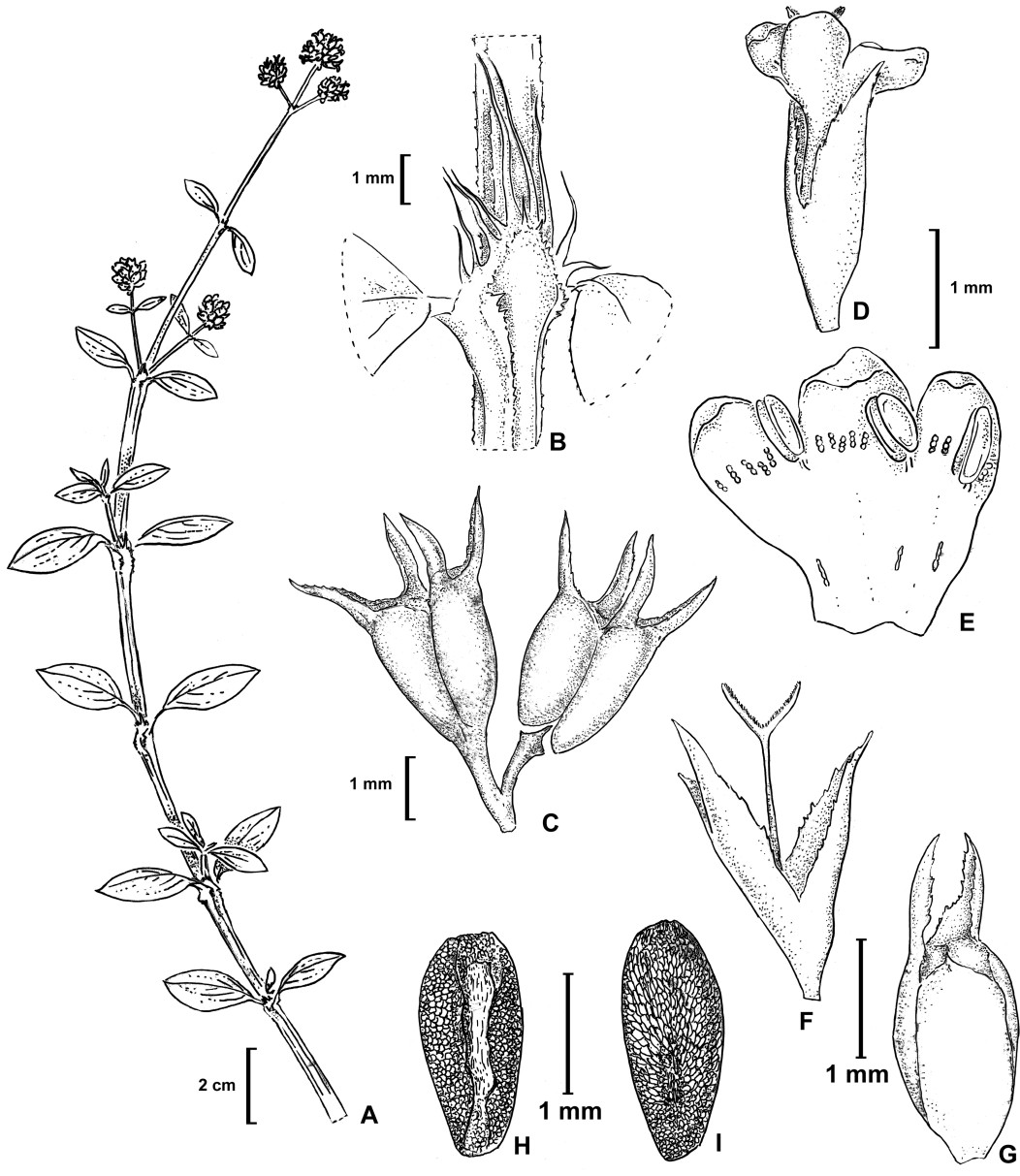

**Figure 2** *Galianthe vasquezii.* (A) Apical part of flowering branch. (B) Stipular sheath. (C) Fruit. (D–F) Flower. (E) Inside of corolla. (F) Style, stigma and calyx. (G) Ventral view of indehiscent valve, calyx tube and lobes. (H–I) Seeds. (H) Ventral view. (I) Dorsal. All from isotype (MO).

remains temporary attached in upper half, at maturity caduceus, seed 1.8–2 × 0.8–1 mm, ovoid, ventral face longitudinally furrowed, partially covered by the strophiole; exotesta reticulate-foveate. Figures 2 and 3. LSID: 77166460-1—*Galianthe vasquezii*.

**Distribution**—Andes of Peru and Colombia

**Observations**—All specimens of the *G. vasquezii* were previously identified as *Galianthe palustris*. However, it differs from *G. palustris,* in having calyx lobes 1–1.4 mm long, with

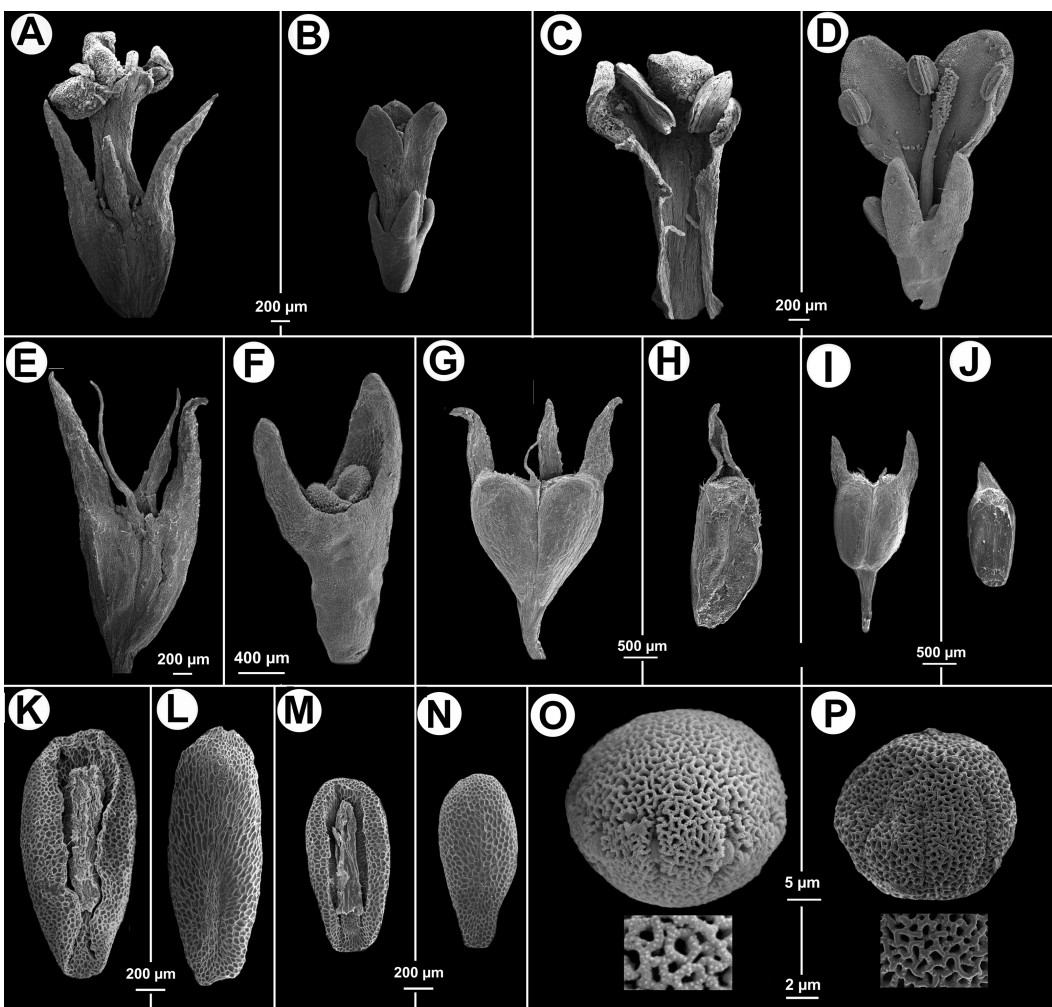

**Figure 3** **Morphological characters distinguishing.** *Galianthe vasquezii* (pictures A, C, E, G, H, K, L, O from the isotype at MO) and *G. palustris* (pictures B, D, F, I, J, M, N, P from A. A. Cabaña 19 at CTES). (A) Flower. (C) Inside the corolla with two stamens. (E) Hypanthium and dimorphic calyx lobes. (G) Entire fruit. (H) indehiscent valve. (K) Ventral face of seed. (L) Dorsal face of seed. (O) Equatorial view of pollen grains, with simple, below a detail of exine. *Galianthe palustris.* (B) Flower. (D) Opened flower showing inside of corolla and three stamens, style and stigma. (F) Hypanthium, nectariferous disc, and calyx lobes. (I) Entire fruit. (J) Indehiscent valve. (M) Ventral face of seed. (N) Dorsal face of seed. (P) Pollen with double reticulum, below a detail of exine showing the suprareticulum incomplete.

acute apex (vs. calyx lobes 0.4–0.6 mm long, obtuse), corolla 1.75–2.1 mm long, lobes internally with hairs scattered at base, tube internally with some dispersed hairs near its base (vs. corolla 1–1.5 mm long, internally glabrous), pollen grains with reticulate exine, muri nanospinose (vs. pollen grains with bireticulate exine, suprareticulum psilate and incomplete, infrareticulum nanospinose), fruit 1.8–2 mm long, deltoid in outline, acropetally dehiscent (vs. fruit 1.1–1.5 mm long, oblong or obovate in outline, basipetally dehiscent), and seeds 1.8–2 mm long (vs. seeds 1–1.42 mm long).

*Ecology*—*Galianthe vasquezii* grows in Montane Forest of Peru and Colombia, which represents a severely fragmented type of vegetation. It grows between 1,800 and 2,500 m altitude.

*Conservation status*—The extent of occurrence (EOO) was calculated to be 397 km$^2$ (cell sized 2 km). Following the IUCN criteria (*IUCN, 2014*), this species should be considered endangered [EN B1 ab (ii, iii)] due to the continuing decline in area and quality of its habitat.

*Additional Specimens Examined*—COLOMBIA: Antioquia, Monte del Diablo, 21 Jul. 1944, *Bro. Daniel 3303* (US); Rio Negro, 16 Dec. 1933, *Bro. Daniel 165* (US). PERU: Distrito Huancabamba, sector Grapanazú, límite Parque Nacional, Yanachaga-Chemillen, 10°26′S, 75°23′W, 15 Oct. 2003, *R. Rojas et al. 1892* (MO); idem, sector Tunqui, camino hacia María Puñis, 1,895 m, 10°16′31″S, 75°30′59″W, *M. Cueva 193* (HOXA, HUT, MO, USM); Luispicanchi, Cuzco, Quincemil, 13°14′S, 70°45′W, Oct. 1950, *F. Marín 2731* (CTES, LIL); Pasco, Oxapampa, carretera de Cochabamba, 10°33′42″S, 75°27′23″W, 11 Nov. 2004, *A. Monteagudo et al. 7587* (CTES, MO).

## New combinations

***Galianthe palustris*** (Cham. & Schltdl.) Cabaña Fader & E. L. Cabral, *comb. nov. Diodia palustris* Cham. & Schltdl., Linnaea 3: 347. 1828. *Borreria palustris* (Cham. & Schltdl.) Bacigalupo & E. L. Cabral, Hickenia 2: 264. 1998. *Spermacoce palustris* (Cham. & Schltdl.) Delprete, Fl. Il. Catarin. (2): 740. 2005. TYPE: BRAZIL, Santa Catarina, "*Ad fretum St. Catharinae Brasiliae ipsi legimus, in palustribus Brasiliae aequinoctiales*", s. d., *F. Sellow s.n.* (holotype: B destroyed, lectotype here designed PR!).

*Borreria gymnocephala* DC., Prodr. 4: 549. 1830. *Diodia gymnocephala* (DC.) K. Schum., in Martius, Fl. Bras. 6(6): 16. 1888. TYPE: BRAZIL. s. d., *J.P. Pohl s. n.* (holotype: G-DC!).

*Borreria pterophora* C. Presl., Abh. Königl. Böhm. Ges. Wiss. V, 3: 516. 1845. *nov. syn.* TYPE: BRAZIL, Rio Janeiro, s.l., s.d., *Beske s.n.* (holotype: PR!).

*Diodia alata* Nees & Mart., Nova Acta Acad. Caes. Leop. Carol., Wied-Neuwied 12: 12. 1824. *Dasycephala alata* (Nees & Mart.) Benth. & Hook. f. ex B.D. Jacks, Index Kew. 2: 719. 1893. TYPE: BRAZIL, s.l., s.d., *M. Wied s.n.* (holotype BR!; isotypes: LD, LE, W!).

*Diodia microcarpa* K. Schum. ex Glaz., Bull. Soc. Bot. France 56 (Mém. 3d): 361. 1909. TYPE: BRAZIL, "*Brasília*", *A.F.M. Glaziou 18283* (holotype B destroyed, photo F 867!).

*Description*

Herb stoloniferous, sometimes with ascendant stems. Stems quadrangular, angle strongly winged, wing with long and slightly recurved fimbriae, or with scabridous and retrorse papillae, especially near foliar nodes. Leaves subsessile or pseudopetiolate; pseudopetiole 1–4 mm long; blades elliptic or obovate, rarely orbicular, 22–30 mm × 10–18.6 mm, apex obtuse, acute or acuminate, base attenuate, plicate-nervose, with 3–4 secondary nerves, adaxially glabrous or scaberulous, margin scaberulous, abaxially scabridous only on nerves; stipular sheath 3–4 mm long, basally alate, 7–9 fimbriate, fimbriae linear, glabrous, 5–8 mm long. Inflorescences thyrsoid, partial inflorescences congested and multiflorous, sometimes arranged on pleiochasium. Flowers shortly pedicellate; pedicel 0.5–1 mm long; calyx 2(-3)-lobed, hypanthium glabrous, lobes triangular, apex obtuse, succulent, 0.4–0.6 mm long;

corolla 2-3-lobed, infundibuliform, white, 1–1.5 mm long, tube internally and externally glabrous, scarcely papillate on apex of the dorsal face of lobes; stamens 2–3, anthers 0.2–0.34 mm long, oblong, dorsal surface with a bullate connective, immediately above the insertion of the filament; pollen grains 6–7(-8) zonocolporate, oblate-spheroidal, small, $P = 24$ μm, $E = 25.3$ μm, long colpi, endoaperture an endocingulum, exine bireticulate, suprareticulum incomplete, muri psilate, infrareticulum complete, muri nanospinose, 0,15–0,28 μm long; style bifid, 1–1.5 mm long, stigmatic branches 0.2–0.46 mm long, notoriously papillate. Fruit a capsule, which separates from the apex downwards into two indehiscent mericarps, both mericarps remain basally united to the pedicel, tardily deciduous, oblong or ovate in outline, glabrous; seeds 1–1.42 × 0.7–0.8 mm, ovate or obpiriforme in outline, ventral surface with a longitudinal furrow covered by a persistent strophiole; exotesta reticulate-foveate. Figure 3. LSID: 77166461-1—*Galianthe palustris*.

**Distribution**—Brazil (Bahia, Minas Gerais, Paraná, Rio de Janeiro, Rio Grande do Sul, Santa Catarina, and São Paulo), and Argentina, Misiones province.

**Ecology**—*Galianthe palustris* is a heliophilous plant that inhabits in swampy areas near lotic water bodies, especially along main rivers and their tributaries.

**Additional Specimens Examined**—ARGENTINA. Misiones: San Pedro, Parque Provincial Moconá, embarcadero, 7 Mar. 2013, *M. D. Judkevich et al. 46* (CTES); idem, borde de arroyo, 11 Dec. 2011, *L. M. Miguel et al. 19* (CTES). BRAZIL. Bahia: Belmonte, 23 Nov. 1970, *T. S. Santos 1124* (CEPEC); Minas Gerais: Camanducaia, Monte Verde, Estrada Camanducaia, 27 Apr. 2013, *J. A. M. Carmo 125* (UEC); ídem, Monte Verde, 24 Jan. 2013, *J. A. M. Carmo 111* (UEC); ídem, Mata dos Vargas, 22 Mar. 2000, *R. B Torres et al. 1176* (FUEL); Santos Dumont, *s. d.*, *H.L.M. Barreto 11339* (BHCB); São João do Manhuaçu, 19 km S of the intersection of Highway BR-116 & BR-262, just N of the village of São João do Manhuaçu, 27 Mar. 1976, *G. Davidse & W. G. D'Arcy 11434* (SP). Paraná: Mun. Bocaiúva do Sul, Serra da Bocaína, 31 Mar. 2001, *E. Barbosa et al. 654* (CTES, ESA, MBM); Serra de São Luís, BR 277, 19 Jan. 1985, *M.S. Ferrucci et al. 284* (CTES); Fazenda Reserva, 85 Km SW of Guarapuava, on bank of brook near Barbaquá, 17 Mar. 1967, *J. C. Lindeman et al. 4959* (CTES, MBM, NY, UB); Mun. Morretes, Serra Morumbi, picada ao Olimpo, 19 Jan. 1995, *O. S. Ribas et al. 761* (CTES); Mun. Piraquara, Floresta, 9 Mar. 1947, *G. Hatschbach 640* (CTES, LIL); BR-476, 7 Km E de Contendas, 26 Jan. 1985, *A. Krapovickas & C. L. Cristóbal 39632* (CTES); Fazenda de J. Rickli near Turvo, 40 km N of Guarapuava, forest, 9 May. 1967, *J. C. Lindeman et al. 5280* (CTES); Curitiba, 17 May. 2002, *J. Cordeiro 2233* (ESA). Rio Grande do Sul: Barracão, Parque Estadual de Espigão Alto, 1 Mar. 2001, *M. Sobral & J. Larocca s.n.* (FURB); Capivari, Viamão, 15 Mar. 1975, *Porto et al. 1389* (CTES); Esteio, 23 Mar. 1949, *B. Rambo 40638* (LIL); Leopoldo, on Monte Jacaré, 7 Dec. 1948, *B. Rambo 38588* (LIL); Pareci, Prope Montenegro, 31 Mar. 1950, *B. Rambo 46536* (CTES); Porto Alegre, 17 Dec. 1932, *B. Rambo s.n.* (P04541549); idem, Morro da Gloria, 16 Dec. 1931, *B. Rambo 577* (LIL); San Salvador, 14 Mar. 1947, *A. Sehnem 2676* (SI); ídem, 16 Dec. 1933, *B. Rambo 577* (SP); idem, Montenegro, 1 Mar. 1950, *A. Sehnem 4426* (SI); Santana, 6 Apr. 1974, *M.C. Sidia 27* (HAS, CTES). Rio de Janeiro: 17 Km from praça da Parati on road from Parati to Cunha, 26 Apr. 1972, *J. H. kirkbride 1729* (US); Nova Friburgo, 12 Nov. 1890, *A. Glaziou 18283* (P02088844); Petrópolis, vale Bonsucesso, 13 Apr. 1968, *B.D.*

*Sucre 2738* (US); Serra da Mantiqueira, Maciço do Itatiaia, Parque Nacional do Itatiaia, 16 Apr. 1971, *I. Gottsberger et al. 110* (CTES); idem, *I. Gottsberger 110-16471* (CTES); Serra dos Orgãos, 11 Jan. 1905, *G. Gardner 445* (US). Santa Catarina: 6.5 KM NW de Aguas Mornas, caminho a Lourdes, 6 Feb. 1994, *A. Krapovickas et al. 44793* (CTES); Am Wege in del Velha bei Blumenau, Oct.1888, *E. H. G. Ule 1062* (US); Fazenda Farofa, trilha da estrada do meio, 6 Apr. 2007, *R. P. M., Souza 103* (ESA); Pilões, Palhoça, 6 Apr. 1956, *R. Reitz et al. 2997* (US, NY); Santa Terezinha, Urubici, 7 Apr. 2009, *M. Verdi et al. 1905* (IFFSC); São Bento do Sul, Rio Natal. Estrada rumo ao Xikavitska (Salto Seco), 19 Feb. 2011, *F. S. Meyer 982* (UFRN); São Bento do Sul, Trilha do Parque 23 de September, 14 Dec. 2014, *P. Schwirkowski 732* (FPS); Taió, Fazenda Tarumã, 18 Feb. 2010, *A. Korte & A. Kniess 1821* (FURB); Três Barras, Guruvá, San Francisco do Sul, 7 Nov. 1957, *R. Reitz et al. 5621* (NY, US); Urubici, Santa Terezinha, 7 Apr. 2009, *M. Verdi et al. 1905* (CTES); São Paulo: Barra do Turvo, 24 Mar. 2005, *M. Carboni, 110* (ESA); 10 km de Barra do Turvo em direção a Pariquera-Açu, 14 Feb. 1995, *J. P. Souza, et al. 96* (SP); Boracéia, 26 Mar. 1940, *N.G. Blanco s. n.* (SP); Campinas, Lago próximo ao parque ecológico da UNICAMP, 1 Jun. 1995, *L. Y. S Aona & A. D. Faria 95/50* (SP); Campos das Sete Lagôas, Fazenda Campininha, just north of Rio Mogi-Guaçu 1,8 km NW of Pádua Sales, Mogi Guaçu, 4 Dec. 1961, *G. Eiten 3517* (SP); Cananéia, Serra do Tambor, Vale do Ribeira, sul do Estado de São Paulo, 20 Nov. 2006, *M. A Pinho-Ferreira et al. 673* (UEC); Cunha, Trilha do Rio Bonito, Parque Estadual da Serra do Mar, 19 Mar. 1996, *A. Rapini et al. 73* (UEC); Cunha, Parque Estadual da Serra do Mar, Núcleo Cunha, 19 Mar. 1996, *A. Rapini, et al. 73* (SP); Estação Biológica, Alto da Serra, 800–900 m, 6 Mar. 1929, *A. Smith 2076* (BA, NY); Estação Experimental, área nativa, Pariquera-Açu, 2 Apr. 1997, *R. B. Torres et al. 182* (IAC); Eldorado, May. 2012, *A. Oriani, et al. 450* (ESA); Eldorado Paulista, P.E. Jacupiranga, Núcleo, Caverna do Diabo, Ilha da Caverna, 24°38′91″S, 48°23′31″W, 9 Feb. 1995, *Leitão Filho et al. 32980* (UEC); Estação Visconde do Rio Claro, 12 Dec. 1888, *A.C.G.G. Loefgren 1220* (SP); Ilha do Cardoso, Jacareu, forest and mangrove swamp, 8 Sep. 1976, *P.H. Davis et al. 60747* (UEC); Itapetininga, 9 Feb. 1976, *H. F. Leitão Filho et al. 1630* (UEC); Itirapina, Ipiranga, 23 Mar. 1906, *A. Usteri s.n.* (SP); Juquiá Sitio Areia Dourada, 29 Nov. 1994, *K. D. Barreto et al. 3290* (CTES); Paranapiacaba, 16 Jun. 1966, *T.M. Pedersen 7795* (CTES, SI); Parque Estadual da Serra do Mar, Núcleo Curucutu, 13 Apr. 2001, *L. D. Meireles et al. 151* (UEC); Pinheiros, 8 Jan. 193, *A. Gehrt s.n.* (IAC); Pindamonhangaba, Fazenda São Sebastião do Ribeirão Grande, noroeste do talhão 10, 22 Feb. 1996, *S. A. Nicolau et al. 1051* (SP); Ponta da Praia, 22 Dec. 1938, *E. Guimarães 5* (SP); Rio Claro, 12 Dec 1988, *A.C.G.G. Loefgren 11782* (NY); Santo Amaro, Seminário do Espírito Santo, 20 Mar. 1943, *L. Roth 10317* (IPA); São Francisco Xavier, Caminho para Cachoeira das Couves, 14 Apr. 1995, *J.Y. Tamashiro et al. 902 (* UEC); São José dos Campos, Distrito de São Francisco Xavier, 14 Apr. 1995, *J.Y. Tamashiro et al. 902* (SP); São Miguel Arcanjo, Parque Estadual Carlos Botelho, 20 Mar. 2002, *S. Bortoleto et al. 31* (UEC); São Sebastião, 22 Apr. 2000, *J. P. Souza et al. 3398* (UEC); Tapiraí, Reserva Particular da Votorantim, 26 Mar. 2013, *C. B. Virillo et al. 8* (UEC).

***Taxonomic notes***—Until the present, *Borreria pterophora* has been considered as an imperfectly known but valid name, which is at present day also registered as an endemism

of Rio de Janeiro state, Brazil (BFG, 2015), however the examination of the holotype deposited at PR revealed us that is a new synonym of *Galianthe palustris*.

**Galianthe spicata** (Miq.) Cabaña Fader & Dessein, *comb. nov. Diodia spicata* Miq., Stirp. Surinam. Select. 179–180, t. 52. 1850. *Dasycephala spicata* (Miq.) Benth. & Hook. f. ex B.D. Jacks., Gen. Pl. 2: 144. 1873. *Borreria spicata* (Miq.) Bacigalupo & E.L. Cabral, Opera Bot. Belg. 7: 307. 1996. *Spermacoce spicata* (Miq.) Delprete, J. Bot. Res. Inst. Texas 1(2): 1028. 2007. TYPE: SURINAM: *Sylvarum prope Bergendaal, H.C. Focke s.n.* (holotype U!, isotypes HAL0113849!, K000265575!).

*Diodia denudata* Standl., J. Wash. Acad. Sci. 15(5): 105. 1925. Type: PANAMÁ, on wet stream bank along the Río Tapia, near sea level, 24 Dec 1923, *P. C. Standley 28123* (holotype: US01154022!).

*Description*

Herb or subshrub 80–140 cm alt., erect, stems simple to much branched. Stems quadrangular, fistulose, glabrous, angle weakly alate or without wings, glabrous. Leaves pseudopetiolate, pseudopetiole 0.5–2 mm long, blades elliptic or narrowly elliptic, papery or subcoriaceous when dry, adaxially glabrescent or scabridous, abaxially glabrous, only scabridous on nerves, base acute or cuneate, apex acute or acuminate, $30–110 \times 10–33$ mm; 5–7 secondary nerves, visible on both faces; stipular sheath 1.5–3 mm long, margin truncate or scarcely triangular, pilose, with 5–7 fimbriae, fimbriae 2–7 mm long, with some antrorse hairs. Inflorescences spiciform, partial inflorescences glomeriform, axillary, (5)10–25 per flowering branch, with 3–20 flowers, bracts foliaceous, decreasing in size towards the apex, sometimes up to the same size than the partial inflorescence. Flowers homostylous, calyx 4-lobed, hypanthium obconic, 0.55–6 mm long, puberulous, lobes 0.2–0.3 mm long, unequal, subtriangular, apex acute, margin ciliate; corolla subtubular, slightly expanded to the apex, 1–1.2 mm long, white or greenish white, sometimes with apex of lobes lilac, glabrescent outside, with a ring of moniliform hairs near insertion of the filaments, tube 0.5–0.7 mm long, lobes ovate, apex acute, internally with some scattered hairs, externally pilose and papillose, especially at the apex, 0.2–0.5 mm long, anthers 0.2–0.25 mm long, oblong, sometimes with a theca visibly smaller than the other, filament 0.15 mm long; pollen grains 7-zonocolpate, prolato-spheroidal, small, $P = 30.3$ μm, $E = 28$ μm, colpi long, endoaperture an endocingulum, tectum bireticulate, microreticulate, suprareticulum psilate, infrareticulum with muri nanospinose or psilate; stigma bifid, divided up to the half of its length, with papillae only in the internal face of the stigmatic branches, scarcely exerted. Fruit a capsule, $1.2–1.4 \times 1–1.2$ mm, longitudinally separated from the pedicel upwards up to median portion of the fruit, both mericarps remain attached to each other at the upper part, mericarps indehiscent, subglobose, ventral face flat, slightly laterally compressed, dorsal face pubescent, hispidulous or glabrescent; seeds $0.8–1 \times 0.35–0.45$ mm, oblong or ovate in outline, plane-convex, ventral face with a ample groove, dark brown or nigrescent; exotesta reticulate-foveate, cells polygonal, almost isodiametric. Figures 4 and 5. LSID: 77166462-1—*Galianthe spicata*.

*Distribution*—Brazil (Amazonas, Roraima, Rondônia, Para, Mato Grosso), Colombia (Chocó), French Guaina, Panamá, Suriname, and Venezuela (Amazonas y Anzoátegui).

*Ecology*—*Galianthe spicata* grows inside or edges of humid forests.

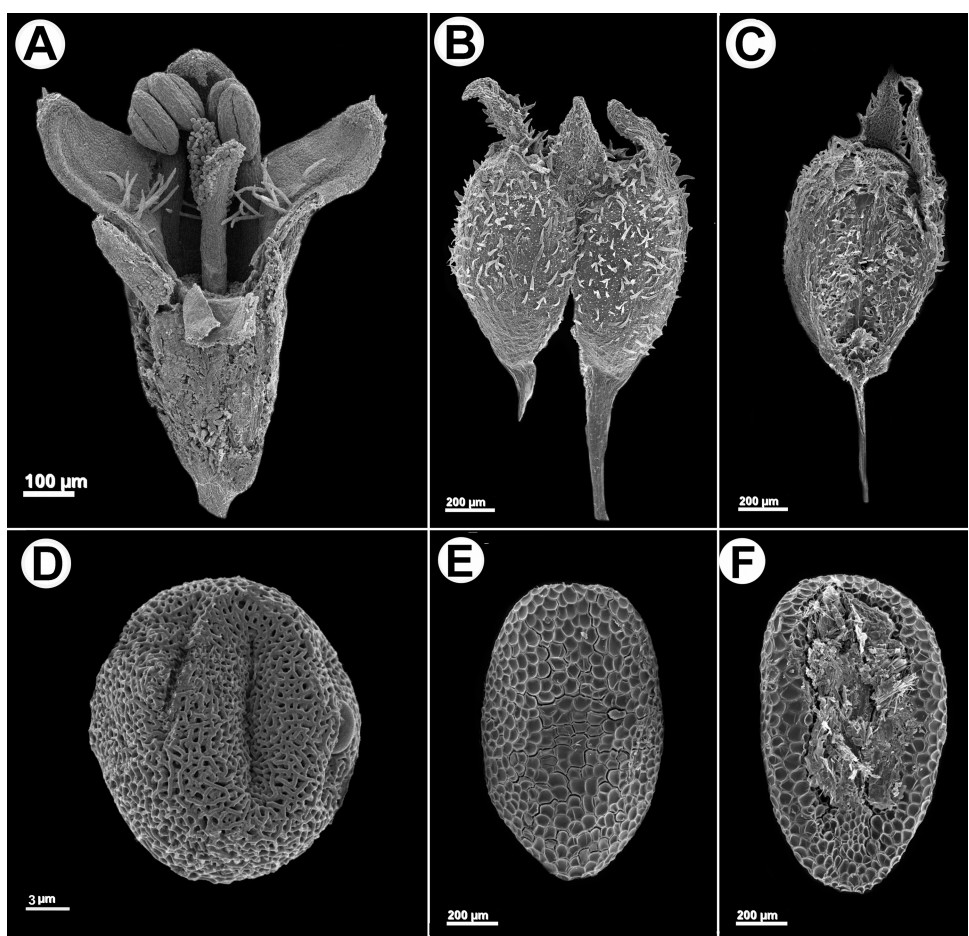

**Figure 4** *Galianthe spicata.* (A) Longitudinal section of flower. (B) Fruit with acropetal dehiscence. (C) Indehiscent valve. (D) Equatorial view of pollen grains showing exine with double reticulum. (E) Dorsal face of seed. (F) Ventral face of seed A: *From P. G. Delprete 11876* (CAY), (B–D): from *M. Sobral et al. 10020* (BHCB).

*Additional Specimens Examined*—BRAZIL: Amazonas, vicinity of Camp Tucano, Rio Tucano, 3 Dec. 1965, *B. Maguire et al. 60319* (IAN, MO); Rondonia, Porto Velho, along hwy 364 92 km, by road NE of junction with, 09°22′S 064°40′W, 20 Apr. 1987, *H. M. Nee 34960* (MO); Roraima, Dormida, Serra do Lua, foothills of Serra da Lua, 13 Jan. 1969, *G. T. Prance 9271* (MO); Pará, Conceição do Araguaia, near Corrego São João and Troncamento Santa Teresa, 8 Feb. 1980, *T. C. Plowman 8524* (MO, NY); Altamira Gleba Curuaé, Jul. 2005. *M. Sobral et al. 10020* (BHCB). COLOMBIA: Chocó, near Madurex Logging Campn above Teresita and below the rapids on Rio Truando, Feb. 1967, *J. A. Duke 9977* (MO); ídem, logging road ca. 2–4 km NW of Teresita, 100 m, 18 May. 1967, *J. A. Duke 11055* (MO). FRENCH GUIANA: Kamakusa, upper Mazaruni River, 23–29 Nov. 1922, *J. S. de la Cruz 2808* (MO); Route de l'Est (N2), Montagne Maripa, c. 31 km S of the Comte bridge, c. selectively logged forest, 04°26′N, 52°20′W, 3 Dec. 1994, *L. Andersson 1961* (MO).

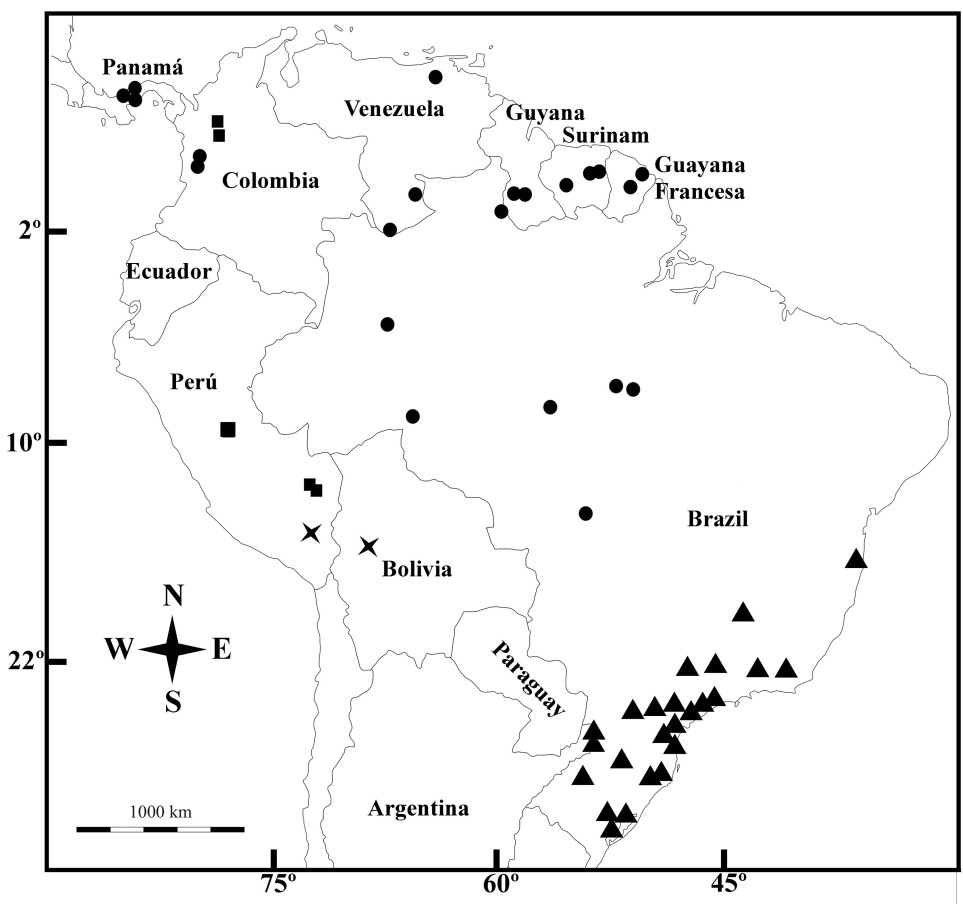

**Figure 5** **Geographic distribution.** *Galianthe boliviana* (X), *G. palustris* (triangle), *G. spicata* (dots) and *G. vasquezii* (square).

GUYANA: Rupununi, Kanuku Mts., Crabwood Cr. Camp 2 forest, on brown loamy sand, 3°07′N, 59°06′W, 260 m, 2 Apr. 1994, *M. J. Jansen-Jacobs et al. 3564* (MO); idem, E Kanuku Mts, NE of Warimure, in forest, 03°05′N 059°20′W, 200–500 m, 23 Jan. 1991, *M. J. Jansen-Jacobs et al. 2189* (MO). PANAMA: Canal Area, Barro Colorado Island, 10–100 m, 9°09′17″N, 79°50′53″W, 16 Dec. 1967, *T. B. Croat 4373* (MO); idem, 9°9′17″N, 79°50′53″W, 18 Mar. 1969, *T. B. Croat 8738* (MO); idem, 9°9′0″N, 79°51′0″W, 1931, *S. Aviles s.n.* (MO); Cerro Azul, 700 m, 9°10′2″N, 79°24′59″W, 29 Jul. 1972, *W.G. D'Arcy 6199* (MO). idem, 700 m, 9°10′13″N, 79°25′13″W, 7 Jun. 1970, *A. Kant 46* (MO); Cerro Jefe, 1,000 m, 9°14′0″N, 79°22′0″W, 12 Sep. 1994, *C. Galdames et al. 1604* (MO); Colón: Santa Rita, 9°20′0″N, 79°47′0″W, 6 Apr. 1969, *W. H. Lewis et al. 5238* (MO); idem, 9°20′0″N, 79°47′0″W, 6 Apr. 1969, *W. H. Lewis et al. 5238* (MO); idem, 9°20′0″N, 79°47′0″W, 6 Apr. 1969, *W. H. Lewis et al. 5238* (MO); idem, 9°20′13″N 079°46′04″W, 31 Jan. 1971, *T. B. Croat 13191* (MO); idem, 9°19′42″N, 79°47′27″W, 9 Jul. 1971, *T. B. Croat & M. P. Duncan 15339* (MO); Gamboa Pipeline Road, 90 m, 9°9′36″N, 79°44′44″W, 9 Feb. 1974, *M. H. Nee 9577* (MO); Pipeline Road, 50–100 m, 9°10′0″N, 79°46′0″W, 23 May. 1969, *R.*

*L. Lazor 3464* (MO). SURINAME: 1850, *F.A.W. Miquel s.n.* (K000265073). Brokopondo: Brownsberg Nature Park, Trail to Mazaroni Val. Primary forest, 04°56′N, 55°11′W, 400–450 m, 24 Jan. 1999, *P.G. Delprete 7083* (MO); Marowijne: Nassau Mts, Plataeu C, lateritic rocky soil, 4°49′N, 54°36′W, 500–550 m, 26 Jan. 2003, *M.J. Jansen-Jacobs et al. 6254* (MO). VENEZUELA: Amazonas, Atabapo, 5 km al Norte de la desembocadura del Rio Orinoco, 2°24′N, 64°24′W, 400 m, Oct. 1991, *E. Marín 1678* (MO); Anzoátegui, Cabeceras del Morichas Largo, entre Santa Elena y San Pedro a unos 30 km Sur de la Viuda, 9 Nov. 1984, *R. A. Montes 2524* (MO).

## First record from Peru

*Galianthe boliviana* E.L. Cabral, Brittonia 57(2): 142, f. 1. 2005. TYPE. BOLIVIA: La Paz: Inqusivi, Cerro Aguada, 2,500–2,800 m, 22 Nov 1991, *M. Lewis 40645* (holotype: LPB0000936!; isotype MO!).

*Distribution*— Sub-Andean foothills of Bolivia and Peru. *Galianthe boliviana* grows mainly on eroded slopes between 1,800 and 3,150 m of altitude, especially in open forest of *Alnus acuminata* Kunth. of Yungas at 2,800–3,000 m of altitude.

*Taxonomic notes*—*Galianthe boliviana* is similar to *Galianthe dichasia* and *G. cymosa* in having cymoidal inflorescences with subglomeriform partial inflorescence, but differs from these in possessing stems 20–30 cm tall, with smooth, glabrous, and narrowly winged angles.

*Additional Specimens Examined*—PERU: Cusco, Quispicanchis, Marcapata, 176 km from Cusco on road to Maldonado, Marcapata to Cocha, 8 Mar. 1991, 13°25′S 070°54′W, 3,150 m, *Percy Núñez V. & C. Paycarmayta 13140* (MO).

Key to *Galianthe* species with indehiscent mericarps (modified from *Cabral & Bacigalupo, 1997*)

| | | |
|---|---|---|
| 1. | Stipular sheath tubular, prolonged above the insertion of the corresponding pairs of leaves............................................................................................................... | 2 |
| 1′. | Stipular sheath truncate, never surpassing the insertion of the corresponding pairs of leaves............................................................................................................... | 3 |
| 2. | Stipular sheath pilose; stems with strongly alate angles; Brazil.................................... ............................................................................ *G. vaginata* E.L. Cabral & Bacigalupo | |
| 2′. | Stipular sheath glabrous; stems without wings. Brazil.................................................. ...... ......................................................... *G. polygonoides* E.L. Cabral & Bacigalupo | |
| 3. | Leaves only with one nerve visible on abaxial face.................................................... | 4 |
| 3′. | Leaves plicate nervose.............................................................................................. | 7 |
| 4. | Inflorescences pauciflorous, in lax cymoid, partial inflorescences 1-florous. Ecuador, Perù.......................................................................................... *G. dichotoma* | |
| 4′. | Inflorescences multiflorous, partial inflorescences multiflorous, in fascicles or glomeruli.............................................................................................................. | 5 |

5. Inflorescences cymoid, partial inflorescences glomeriform, calyx 4-lobed, pollen with reticulate exine. Colombia........................................................................
......................................................... *G. bogotensis* (Kunth) E. L. Cabral & Bacigalupo

5′. Inflorescences thyrsoid, spiciform, or cymoid, partial inflorescences fasciculate, calyx 2-4-lobado, pollen grains with bireticulate exine............................................. 6

6. Inflorescences thyrsoid-spiciform or cymoid, primary axis shorter than the laterals, calyx 2 (-4) lobed, corolla of long-styled flowers with a fringe of hairs from apex of anthers to base of tube; pollen grains 6–7 zonocolpate, both reticula complete, fruit 2–3 times wider than long. Argentina, SE, and S Brazil, Paraguay, and Uruguay.....
......................................................... *G. brasiliensis* (Spreng.) E.L. Cabral & Bacigalupo

6′. Inflorescences thyrsoid-spiciform, primary axis longer than the laterals, calyx always 4-lobed, corolla of the long-styled flowers with ring of hairs, pollen grains 8–10 zonocolpate, suprareticulum incomplete, fruit as long as wide, , Mesoamerica
......................................................... *G. angulata* (Benth.) Borhidi

7. Partial inflorescences congested, glomeriform or subglomeriform........................... 8

7′. Partial inflorescences pauciflorous, fasciculate......................................................... 13

8. Inflorescences spiciform, con 5–20 partial inflorescences per flowering branch. Brazil, Colombia, French Guiana, Panama, Surinam, and Venezuela.......................
......................................................... *G. spicata*

8′. Inflorescences thyrsoid or cymoid, with 3–5 partial inflorescences per flowering
......................................................... 9

9. Stems notoriously alate.......................................................................................... 10

9′. Stems obscurely alate............................................................................................. 12

10. Inflorescences thyrsoid or with a simple axis, partial inflorescences glomeriform, flowers homostylous, calyx 2–3 lobed, corolla 2–3 lobed....................................... 11

10′. Inflorescences cymoid, partial inflorescences subglomeriform, flowers distylous, calyx 4-lobed, corolla 4-lobed, Argentina, Brazil, Paraguay, and Uruguay.....................
......................................................... *G. dichasia* (Sucre & C.G. Costa) E.L. Cabral

11. Calyx lobes 1–1.4 mm long, with acute apex, corolla 1.75–2.1 mm long, corolla lobes internally with hairs scattered at base, tube internally with some dispersed hairs near its base, pollen grains with reticulate exine, muri nanospinose, fruit 1.8–2. mm long, deltoid in outline, acropetally dehiscent, seeds 1.8–2 mm long..............
......................................................... *G. vasquezii* R.M. Salas & J. Florentín

11′. Calyx lobes 0.4–0.6 mm long, obtuse, corolla 1–1.5 mm long, internally glabrous, pollen grains with bireticulate exine, suprareticulum psilate and incomplete, infrareticulum nanospinose, fruit 1.1–1.5 mm long, oblong or obovate in outline, basipetally dehiscent, seeds 1–1.42 mm long.................................................................
......................................................... *G. palustris* (Cham. & Schltdl.) Cabaña Fader & E. L. Cabral

12. Stems retrorse-scabridous on angles, leaves 1–7 mm lat. Brazil.................................
......................................................... *G. cymosa* (Cham.) E.L. Cabral & Bacigalupo

12′. Stems glabrous, leaves 7–12 mm lat. Bolivia and Perú................................................
......................................................... *G. boliviana* E.L. Cabral

13. Stems scarcely branched; fruit sub-hemispherical, 1.6–2 mm long; Brazil, Paraguay, and Argentina.................... *G. hispidula* (A. Rich. ex DC.) E.L. Cabral & Bacigalupo

13′. Stems much branched; fruit turbinate, 5 mm long; Brazil..........................................
.......................................................................... *G. humilis* E.L. Cabral & Bacigalupo

## DISCUSSION

*Galianthe palustris* and *G. spicata* share the same taxonomic and nomenclatural history. First, they were described under *Diodia*, later they were added to genus *Borreria* (*Bacigalupo & Cabral, 1996*; *Bacigalupo & Cabral, 1998*) due to the presence of homostylous flowers and type of fruit. Later they were, transferred to the genus *Spermacoce* (*Delprete, Smith & Klein, 2005*; *Delprete, 2007*). In 1998, *Bacigalupo & Cabral (1998)* observed that *G. palustris* (then still *Borreria palustris*) is characterized by a thyrsoid inflorescence that is similar to that of *Galianthe*. Despite this remarkable observation, the authors decided to transfer the species to genus *Borreria*. Nearly a decade later, *Delprete, Smith & Klein (2005)* and *Delprete (2007)* transferred both species to *Spermacoce* in an attempt to create a broad genus concept for *Spermacoce*.

Despite overall molecular evidence, *Galianthe spicata* and *G. palustris* also share similar morphological characteristics with the other *Galianthe* species (e.g., spiciform and thyrsoid inflorescences, a bifid stigma and pollen grains with a double reticulum). This last character appears in most species of *Galianthe,* except for *G. bogotensis* (Kunth) E.L. Cabral & Bacigalupo, *G. dichotoma* (Willd. ex Roem. & Schult.) E. L. Cabral & Bacigalupo, and the new species *G. vasquezii,* which have simple reticulum. *Pire (1997)* hypothesized that in a genus mainly represented by species with double reticulum pollen grains, the simple reticulum is the result of the absence of an infrareticulum persisting only a suprareticulum.

Current molecular data indicates that the phylogenetic position of *Diodia palustris* (*Galianthe palustris*) and *D. spicata* (*G. spicata)* make *Galianthe* paraphyletic. The *Galianthe* clade, including both former *Diodia* species, is strongly supported and has two molecularly well-defined clades. The (*Diodia palustris* + *D. spicata*) + *G. brasiliensis* clade is composed only by species with capsules separating into two indehiscent mericarps and which is a diagnostic character of *Galianthe* subgen. *Ebelia*. The sister clade, [*G. eupatorioides* + *G. grandifolia* ] + *G. peruviana*, includes species of *Galianthe* subgen. *Galianthe,* and is characterized by fruits with dehiscent valves. Both morphological and molecular data support the transfer of two former *Diodia* species to *Galianthe*, and more specifically in subgen. *Ebelia*. Additionally, and according to present sampling, the two subgenera described by *Cabral & Bacigalupo (1997)* seem to be monophyletic. The transfer of *Diodia spicata* to *Galianthe* was originally proposed by *Dessein (2003)*, based on fruit, polynological and molecular features.

Even though morphological and molecular data show that three species share several characteristics with *Galianthe* subgen. *Ebelia*, there is a significant difference with the other species of the subgenus. The three species, unlike the remainder, have homostylous flowers. As a result, these results demonstrate the presence of a new floral trait in *Galianthe* and therefore strongly modify the generic concept of the genus.

According to *Groeninckx et al. (2009)*, distyly is often related with double reticulum pollen grains in the tribe Spermacoce. Nevertheless, in the genus *Galianthe* there are some exceptions to this generalization (e.g., *G. bogotensis* (distyly and simple reticulum), *G. spicata* and *G. vasquezii* (homostyly and double reticulum), and *G. palustris* (homostyly and simple reticulum). *Cabral & Bacigalupo (1997)* mentioned that *G. dichotoma* presents an intermediate state between distyly/homostyly and pollen with simple reticulum. The authors defined this phenomenon as an "unclear dimorphism" (in Spanish "dimorfismo poco manifiesto"). Future studies are necessary in order to clearly define the floral morphs that are present in these species.

## ACKNOWLEDGEMENTS

We thank the herbarium curators for providing material, especially James Solomon from MO and Rocio Rojas from HOXA. We also thank HOXA's staff; Rodolfo Vasquez and Thania Carhuaricra for sending us images of *Galianthe vasquezii*'s. We also thank Pedro Cuaranta for his help in the illustration of the new species. The second author thanks Beatriz Galati for the selfless assistance in the observation in MEB (UBA) of the pollen grains, and reproductive structures of *Galianthe palustris* and *G. spicata*. The third author thanks Charlotte Taylor for the invaluable collaboration.

## APPENDIX

List of taxa used in the molecular phylogenetic analysis with voucher information (geographical origin, collector, collector number, herbarium, ITS and ETS accession number) and GenBank accession numbers.

Ingroup. *Borreria* G. Mey. *B. alata* (Aubl.) DC., Brazil, Goiás, Queiroz et al. 14105 (CTES, HUEFS; KF736995, KF737036); *B. capitata* (Ruiz & Pav.) DC., Brazil, Bahia, Queiroz et al. 13688 (CTES, HUEFS; KF736989, KF737031). *B. dasycephala* (Cham. & Schltdl.) Bacigalupo & E.L. Cabral, Argentina, Misiones, Salas & Cabaña 388 (CTES; ITS KF73699); *B. multibracteata* E.L. Cabral & Bacigalupo. Brazil, Goiás, Queiroz et al. 14261 (CTES, HUEFS; KF736990, KF737032); *B. latifolia* (Aubl.) K. Schum., Brazil, Goiás, Queiroz et al. 14110 (CTES, HUEFS; KF736994, KF737035); *B. schumannii* (Standl. ex Bacigalupo) E.L. Cabral & Sobrado, Argentina, Misiones, Cabral et al. 760 (CTES; KF736997, KF737038); *B. tenella* (Kunth) Cham. & Schltdl., Brazil, Queiroz et al. 14252 (CTES, HUEFS; KF736988, KF737030); *B. verticillata* (L.) G. Mey., Argentina, Corrientes, Salas 402 (CTES; KF736998, KF737039); *Carajasia, C. cangae*, Brazil, Pará, Costa et al. 588 (BHCB; KF737015, KF737057); Giorni et al. 179 (BHCB; KF737016, KF737058). *Crusea* Cham. & Schltdl., *C. calocephala* DC., Mexico, Oaxaca, Ochoterena et al. 456 (BR; KF737009, KF737051); *C. coccinea* DC., Mexico, Oaxaca, Ochoterena et al. 461 (BR; KF737010, KF737052). *Diodia* L. *D. saponariifolia* Cham. & Schltdl., Argentina, Misiones, Cabaña & Salas 22 (CTES; KF737007, KF737049). *D. virginiana* L., USA, Missouri, Taylor 12758 (MO; KF737008, KF737050). *Emmeorhiza* Pohl ex Endl. *E. umbellata* (Spreng.) K. Schum., Brazil, Bahia, Queiroz et al. 13746 (CTES, HUEFS; KF737000; KF737042). *Ernodea* Sw. *E. littoralis* Sw., Cuba, Habana, Rova et al. 2286 (GB; KF737001, KF737043).

*E. taylori* Britton, North Bimini, Correll 44186 (NY; KF737002, KF737044). *Galianthe* Griseb. *G. brasiliensis* (Spreng.) E.L. Cabral & Bacigalupo, Argentina, Misiones, Cabral et al. 758 (CTES; KF737011, KF737053). *G. eupatorioides* (Cham. & Schltdl.) E.L. Cabral, Brazil, Goiás, Queiroz et al. 14190 (CTES, HUEFS; KF737012, KF737054). *G. grandifolia* E.L. Cabral, Brazil, Distrito Federal, Queiroz et al. 14015 (CTES, HUEFS; KF737013, KF737055). *G. peruviana* (Pers.) E.L. Cabral, Brazil, Minas Gerais, Belo Horizonte, Salas et al. 408 (BHCB, CTES; KF737014, KF737056). *G. palustris* (Cham. & Schltdl.) Cabaña Fader & E. L. Cabral, Verdi et al. 1905 (CTES; MF166824, MF166826); Miguel et al. 19 (CTES; MF166825, MF166827); *G. spicata* (Miq.) Cabaña Fader & Dessein, Brazil, French Guiana, Andersson et al. 1961 (GB; AM939535, AM933008); *Hexasepalum* Bartl. ex DC. *H. apiculatum* (Willd.) Delprete & J.H. Kirkbr., Brazil, Bahia. Queiroz et al. 13727 (CTES, HUEFS; KF737003, KF737045). *H. angustifolium* Bartl. ex DC., Mexico, Rzedowski 17792 (NY; KF737004, KF737046). *H. sarmentosum* (Sw.) Delprete & J.H. Kirkbr., Cameroon, Dessein et al. 1521 (BR; KF737005, KF737047). *H. teres* (Walter) J.H. Kirkbr., Brazil, Goiás, Queiroz et al. 14089 (CTES, HUEFS; KF737048, KF737006). *Mitracarpus* Zucc. *M. carnosus* Borhidi & Lozada-Pérez, Mexico, Oaxaca, Ochoterena et al. 516 (BR; KF736999, KF737040). *M. megapotamicus* (Spreng.) Kuntze, Argentina, Corrientes, Salas & Cabaña 399 (CTES; ETS KF737041). *Psyllocarpus* Mart. & Zucc. *P. asparagoides* Mart. ex Mart. & Zucc., Brazil, Minas Gerais, Itacambira, Salas et al. 411 (BHCB, CTES; KF737018, KF737060). *P. phyllocephallus* K. Schum., Brasil, Distrito Federal, Queiroz & al. 14016 (CTES; ETS KF737061). *Richardia* L. *R. grandiflora* (Cham. & Schltdl.) Steud., Brazil, Bahia, Nova Roma, Queiroz et al. 14055 (CTES, HUEFS; KF737027, KF737066). *R. humistrata* (Cham. & Schltdl.) Steud., Argentina, Misiones, Bernardo de Irigoyen, Cabaña & Salas 17 (CTES; KF737028, KF737067). *Schwendenera* K. Schum. *S. tetrapyxis* K. Schum., Brazil, Paraná, Marques et al. 83 (CTES; KF737017, KF737059). *Spermacoce* L. *S. breviflora* F. Muell ex Benth., Australia, Harwood 1070 (BR; KF737019, KF737062). *S. confuse* Rendle, Mexico, Ochoterena et al. 552 (BR; KF737020, KF737063). *S. dibrachiata* Oliv., Zambia, Dessein et al. 626 (BR; ITS KF737021). *S. eryngioides* (Cham. & Schltdl.) Kuntze., Argentina, Salas et al. 378 (CTES; KF736992, KF737033). *S. glabra* Michx., USA, Missouri, Perry, Taylor 12757 (MO; KF737022, KF73706). *S. incognita* (E.L. Cabral) Delprete., Brazil, Goiás, Queiroz et al. 14049 (CTES, HUEFS; KF736993, KF737034); *S. prostrata* Aubl., Brazil, Goiás, Nova Roma, Queiroz et al. 14083 (CTES, CTES; KF736996, KF737037); *S. tenuior* L., México, Novelo et al. s/n (BR; KF737023, KF737065). *Staelia* Cham. & Schltdl. *S. herzogii* (S. Moore) R.M. Salas & E.L. Cabral, Bolivia, Santa Cruz, Soto et al. 1053 (CTES, USZ; ITS KF737024). *S. virgata* (Link ex Roem. & Schult.) K. Schum., Brazil, Bahia, Salas et al. 423 (CTES, HUEFS; ITS KF737025); Brasil, Piauí, Salas et al. 443 (CTES, HUEFS; ITS KF737026). Outgroup. *Bouvardia* Salisb. *B. ternifolia* (Cav.) Schltdl., Mexico, Oaxaca, Ochoterena et al. 454 (BR; KF736987, KF737029).

### Funding

This work was supported by a doctoral grant of CONICET Argentina. The funders had no role in study design, data collection and analysis, decision to publish, or preparation of the manuscript.

### Grant Disclosures

The following grant information was disclosed by the authors:
CONICET Argentina.

### Competing Interests

The authors declare there are no competing interests.

### Author Contributions

- Javier Elias Florentín, Andrea Alejandra Cabaña Fader, Roberto Manuel Salas and Steven Dessein conceived and designed the experiments, performed the experiments, analyzed the data, contributed reagents/materials/analysis tools, wrote the paper, prepared figures and/or tables, reviewed drafts of the paper.
- Steven Janssens performed the experiments, analyzed the data, contributed reagents/materials/analysis tools, wrote the paper, reviewed drafts of the paper.
- Elsa Leonor Cabral reviewed drafts of the paper.

### Data Availability

Sequences obtained in this study were deposited at GenBank [*Diodia palusttis*, Vetdi et al. 1905, ETS (MF166824), ITS (MF166826); Miguel et al. 19, ETS (MF166825), ITS (MF166827).

### New Species Registration

The following information was supplied regarding the registration of a newly described species:

77166461-1 – *Galianthe palustris*
77166462-1 – *Galianthe spicata*
77166460-1 – *Galianthe vasquezii.*

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
