# Peer review of "Morphological and molecular data confirm the transfer of homostylous species in the typically distylous genus Galianthe (Rubiaceae), and the description of the new species Galianthe vasquezii from Peru and Colombia"

_PeerJ, doi:10.7717/peerj.4012_

## Round 0.1 · original submission · Major Revisions

Dear Javier Elias,

Please follow the reviewers' recommendations and submit a new and improved version of your manuscript that we could consider for publication in PeerJ,

Cheers,

Marcial.

Reviewer 1 ·

Basic reporting

The writing is usually clear (but the discussion and part of the results should be rewritten) and there is strong evidences (molecular and morphological) supporting the taxonomical work. The literature is cited purposefully. However, the text sometimes overwhelmed the reader with useless details (see e.g. M&M section), while the figures have been rather sloppily prepared (see comments below).

Some subheadings should be modified to reflect their content and the result section should be rearranged (see details comments below).
Furthermore, the text can be simplified and shortened in several places (see some suggestions below).

Part of the results should be rewritten and I will suggest removing the chapter discussion that is unnecessary (as this MS is a taxonomical treatment).

Experimental design

This MS deals with the taxonomy of three plant species, there is no research questions or experimental design to evaluate.

Validity of the findings

no comment

Additional comments

Review of MS for PEERJ – Manuscript ID 18193 – Florentin et al., Morphological and molecular data confirm homostyly in the mainly distylous genus Galianther (Rubiaceae).

The 28th of June 2017

This manuscript deals with taxonomic changes among the “Spermacoce clade” of the tribe Spermacoceae (Rubiaceae). In particular the authors transfer two species Diodia spicata and D. palustris to the genus Galianthe and describe a new species Galianthe vasquezii R.M.Salas & E.L.Cabral from Peru.
The work is mainly based on results found in previous study (i.e. Dessein 2003 and Groeninckx et al. 2009). It provides four new DNA sequences for the species Diodia palustris and uses herbarium-based morphological observations.

The writing is usually clear (but the discussion and part of the results should be rewritten) and there is strong evidences (molecular and morphological) supporting the taxonomical work. The literature is cited purposefully. However, the text sometimes overwhelmed the reader with useless details (see e.g. M&M section), while the figures and part of the text have been rather sloppily prepared (see comments below).

As stated in the review guidelines, I will not comment on the impact and novelty of this study. As this is a MS dealing with the taxonomy of three plant species, there is no research questions or experimental design to evaluate.

It is yet unclear to me if taxonomical treatments are considered as “Research Article” by PEERJ. If yes, my impression is that this taxonomical note should be considered as worth to be published.

However, in my opinion, there are some issues to be addressed before publishing this study.
The authors emphasize, in the MS title, the presence of homostylous species within a mainly distylous genus. But do not discuss the evolutionary implications. They rather treat this biologically interesting phenomenon (see Darwin 1877 book on the different forms of flowers on plants of the same species or the Webb & Lolyd 1986 paper in New Zealand Journal of Botany 24: 163-178 and many others) as a character state.

Therefore, the title does not reflect the work done. I will suggest two options:
The first one: to adapt the title to reflect the work (e.g. Galianthe vasquezii, a new Rubiaceae species from Peru and the transfer of two Diodia species to Galianthe)
The second one: to use the current phylogenetical framework (i.e. the one published by Groeninckx et al. 2009 and the authors’ addition of D. palustre) to discuss the evolutionary implication of homostylous species within a distylous clade. In that case the author should be aware that with only 6 species (out-of the 50 of the genus) their discussion would remain highly speculative.

Additionally some subheadings should be modified to reflect their content and the result section should be rearranged (see comments below).
Furthermore, the text can be simplified and shortened in several places (see comments below).
Part of the results should be rewritten and I will suggest removing the chapter discussion that is unnecessary (as this MS is a taxonomical treatment).


Suggestions and minor issues:

P2, L15 “The remaining species described …” How many species of Dodia were transferred to the different genera.
P2, L18-20. The sentence is unclear. Do you mean that current species of Galianthe subgen Ebelia are all distylous?
P2, L21-28. I would suggest simplifying the whole paragraph, by acknowledging that you follow the work of Dessein (2003) and as a result move Diodia spicata, along with a recently sequenced species (i.e. D. plaustre) to Galianthe. By doing this you also describe a new species G. vasquezii. (as stated in P2, L30).

P3, L11. Remove “Acronyms according to Index Herbarorium”, as this is obvious.
P3, L11. Change: “were analysed” by “were observed” or “were studied”
P3, L12-13. Are you sure PEERJ allow website citation within the main text? To be checked.
P3 L22-29. Is this necessary? It appears as obvious to me (following the last International Code of Nomenclature for algae, fungi and plants, art 29).

P4, L26-28. I believe you can spare some words here by saying that you “align your sequences with MAFFT and subsequent manual corrections in Geneious”.
P4, L24. change “Data analyses”, by “phylogenetic analyses”

P5, L13. Result section should be reorganized.
The “molecular study” at P15L18 (but I guess you mean “phylogenetic results”) should come first and the “Taxonomic Treatment” second (Because you based your taxonomic treatment on your phylogenetic analyses).
P5, L19. You may want to have the “Description-” subheading here, to be consistent with the following subheading (e.g. Distribution-, Ecology-, Conservation status-, etc.). I cannot recall if a Diagnosis is mandatory for a name to be validly published. You may want to check.
P5, L22. Replace “o” by “or”

P6, L2. What are P= 31 and E =29. You should state to what P and E refer.
P6, L3. “muri nanospinose”, what does this mean? It should be clarified for the readers.
P6, L3. “with notorious papillae” do you mean “conspicuous papillae”?
P6, L6. “caducous.; seed…” remove the “.” (i.e. “caducous; seed…”)
P6, L14. “21 Jul 1944” replace by “21 Jul. 1944”
P6, L15. “15 Dic 1933” replace by “15 Dec. 1933”

Throughout the text, when a month is abbreviated it should be followed by a point.

P6, L23. Change “nov. comb.” by “comb. nov.”

P7, L10-29. Same remarks than above for the subheadings, the P= and E = meanings.
P7, L28. What does “u” means?

P8, L5 to P10, L5. “Additional Specimens Examined” I did not review that part!

P10, L19. Add the subheading “description-”

P11, L5 what does P and E mean?

P11, L17 to P12, L 19. “Additional Specimens Examined” I did not review that part!

P15, L18 to P16, L23. “Molecular Study”: I guess you mean “phylogenetic results” needs to be rewritten in a clearer way. I am lost in your results, which sometimes contradict Figure 4.

A few none-exhaustive comments:
P15, L26. Are the genera unresolved or do you lack of statistical support?
P16, L2, remove the “s” or did you mean “statistical”?
P16, L3 what is a “low supported (40) to unsupported clade”. Is that statistically supported or not? The answer is “statistically not supported”.

I suggest removing the discussion, as this is a taxonomical treatment. Furthermore, It is unclear to me what you are discussing.

Figure 2:
Instead of presenting 2x6 pictures, you may want to present 4x3 pictures.
The caption should be rewritten as follow: Morphological characters distinguishing Galianthe vasquezii (pictures A-F from the Isotype in MO) and G. Palustris (Pictures G-L from A.-A. Cabana 19 in CTES). And then the details of each pictures

Figure 4:
In the caption, replace the “y” by “and”

Figure 5:
In the caption explain what are the numbers above the branches…
I suggest to collapse the branches statistically unsupported. (i.e. <51)

Reviewer 2 ·

Basic reporting

Authors analyzed the taxonomic position of two species that were before assigned to Diodia genus: Diodia spicata and D. palustris; but currently they have been translated to Galianthe genus. They use morphological and molecular data (ETS and ITS). The uncertainty position occurs due to some morphological differences exist between these two species and some Galianthe subgen. Ebelia species, mainly the homostylous flowers. Also authors describe a new species (Galianthe vazquezii) and provide a dichotomous key for taxa with indehiscent mericarps. They conclude that both analyzed species fall among Galianthe species subgen. Ebelia.

The English language of the manuscript is clear and correct. The Intro and background are well supported. Figures are in high quality and are well labeled and describe. However, references along the manuscript don’t seem to follow the PeerJ style (are in cursive font). Some details in the Intro can help to improve the manuscript.

1. In the abstract, and along the manuscript, authors mention the phrase: “generic concept” of Galianthe subgen. Ebelia, however in the introduction they never clarify which is this concept. They just mentioned some traits with taxonomic importance. It could be helpful to extend this concept.
2. In the Intro (Page 2; Line 22-26) authors describe three different taxonomic proposals in order to relocate the taxonomic position of D. spicata and D. palustris. Finally, they decided “to verify the position of…. within Galianthe” genus. It is not clear why they decided to keep the Galianthe hypothesis over the Borreria subgen. Dasycephala hypothesis. Although they mention that species “share most affinities with Galianthe subgen. Ebelia” (Page 2; Line 27), is not clear if the decision is due to is the position currently accepted. They could clarify the point.
3. Authors mention (Page 2; Line 27-28) that studied species have some differences with other species of the subgen. Ebelia, but they don´t mentioned these differences. It could be clarify with the point 1.

Experimental design

The method used was well defined and analyses are generally appropriate. The design of the morphological and molecular study was based on a good sampling of species of Spermacoce clade, however, it is noticeable that there is just one species of Galianthe subgen. Ebelia. (G. brasiliensis). Although this was correct due to this especies is the type of subgen. Ebelia, it is recommended the use of more species. Molecular markers in general showed well resolution. The methods and results are described with sufficient detail. Some suggestions:

1. Page 4; Line 9. Authors mention three accession numbers added, but they just report four of them (Page 4; Line 22-23). Check.
2. Why you don´t use Bayesian inference analyses instead of ML bootstrapping to obtain the posterior probability values as a measure of clade support?.
3. Page 6; Line 3: be consistent with decimal separators (dots or commas).
4. Page 6; Line 6: caducous; (without dot)
5. Page 7; Line 27-28: be consistent with decimal separators (dots or commas).
6. Page 11; Line 5: be consistent with decimal separators (dots or commas).
7. Page 14; Line 6: x=15, (without dot)
8. Page 14; Line 12: branch. Brazil. (space)
9. Page 16; Line 2: an extra “s” Carajasia has moderate.
10. Page 16; Line 7-9: write values of clade support by each minor clade mentioned.
11. Page 16; Line 13-14: write values of clade support of Borreria species from the Americas.
12. Page 16; Line 15-18: Mention values of clade support at least for the two subgroups of Spermacoce.
13. Page 17; Line 15: “in” or “is”

Figure 5. Could be better if also highlight Borreria and Spermacoce clades.

Validity of the findings

The data and the results are robust and improve the knowledge of the Galianthe genus. The conclusions are well stated, linked to the main question and limited to supporting results.

Additional comments

I consider that the manuscript could potentially become acceptable for publication in PeerJ. Corrections could be considered as minor corrections but they can help to make it easier for the reader.

---

## Round 0.2 · Minor Revisions

Dear Javier Elias,

Both reviewers agree that the current version of the manuscript has improved in comparison with the previous one.

Please, include suggestion from reviewer #1 in the manuscript, and submit a new version to be consider for publication in PeerJ.

Cheers,

Marcial.

Reviewer 1 ·

Basic reporting

This is the second review of this manuscript (previously submitted to PEERJ). This MS deals with taxonomic changes among the “Spermacoce clade” of the tribe Spermacoceae (Rubiaceae). In particular the authors transfer two species Diodia spicata and D. palustris to the genus Galianthe and describe a new species Galianthe vasquezii R.M.Salas & E.L.Cabral from Peru.

The writing is clear and the discussion and results section have been improved. I made some comments below.

Before being published, the MS need to be carefully checked by a native English speaker (I am not in a position to correctly assess this part – but my feeling is that there is room for improvement).

As mentioned in my previous review, the authors emphasize, in the MS title, the presence of homostylous species within a mainly distylous genus. It is a strategy that will attract readers and at the same disappoint some of them because of the absence of discussions regarding the evolutive implications of this finding. Nevertheless, as the authors consider distily/homostly as a morphological character state, it is acceptable.

The aims of the MS need some clarifications.

First, in an unclear sentence (P2L16-17; see suggestion below), the authors say “in this work we studied” … either the genus Diodia or the species with indehiscent mericarps (this is unclear to me). Then, to exemplify their focus, they list four species (P2L18 - namely: D. bogotensis, D. brasiliensis, D. cymosa, D. hispidula).

Later in the introduction (P2L27-30), they discuss more specifically the position of two related species (D. spicata and D. palustris).

Then (P3L1-2): “As for Diodia spicata, Dessein (2003) proposed to transfer it to Galianthe based on molecular data, pollen grains (double reticulum), and fruit morphology”. Followed by “the aim of this work is to verify the position of D. palustris and D. spicata within Galianthe based on molecular and morphological data.”

The problem is that Dessein (2003) already did it. Thus (as mentioned in my previous review), I believe that the aim of the paper is rather the taxonomical transfer of D. spicata (along with its close relative D. palustris) to Galianthe. Which, by the way, is a honourable, perfectly publishable, aim.
Now, what about D. bogotensis, D. brasiliensis, D. cymosa, D. hispidula nothing is mention in the M&M or the Result sections about those four species?!?


Additionally, before publishing this MS, I recommend the authors to go through the text again and make clearer that they consider distyly (respectively homostyly) as a morphological character state (see e.g. comments below).
In some instance homostyly or distyly is presented e.g. like a syndrome (P1L28) but without explicitly mentioning a syndrome of what? Or as “scarcely manifested” (P18L22) - what does this mean?

Experimental design

Because this MS deals with the taxonomy of three plant species, there is no research questions or experimental design to evaluate.

Validity of the findings

the data support very well the taxonomical changes made by this paper.

Additional comments

Review of MS for PEERJ – Manuscript ID 18193 – Florentin et al., Morphological and molecular data confirm homostyly in the genus Galianthe (Rubiaceae), based on the new species G. vasquezii from Peru and the two new combinations of Dioda species.

The 9th of September 2017

This is the second review of this manuscript (that have been previously submitted to PEERJ). This MS deals with taxonomic changes among the “Spermacoce clade” of the tribe Spermacoceae (Rubiaceae). In particular the authors transfer two species Diodia spicata and D. palustris to the genus Galianthe and describe a new species Galianthe vasquezii R.M.Salas & E.L.Cabral from Peru.

The writing is clear and the discussion and results section have been improved. I made a few comments below.

The MS need to be carefully checked by a native English speaker (as I am not in a position to correctly assess this – but my feeling is that there is room for improvement)

I am not commenting on the impact and novelty of this study. Because this MS deals with the taxonomy of three plant species, there is no research questions or experimental design to evaluate.

As mentioned in my previous review, the authors emphasize, in the MS title, the presence of homostylous species within a mainly distylous genus. It is a strategy that will attract readers and at the same disappoint some of them because of the absence of discussions regarding the evolutive implications of this finding. Nevertheless, as the authors consider distily/homostly as a morphological character state, it could be acceptable.

The aims of the MS are not clearly presented.

First, in an unclear sentence (P2L16-17; see suggestion below), the authors say “in this work we studied” … either the genus Diodia or the species with indehiscent mericarps (this is unclear to me). Then, to exemplify their focus, they list four species (P2L18 - namely: D. bogotensis, D. brasiliensis, D. cymosa, D. hispidula).

Later in the introduction (P2L27-30), they discuss more specifically the position of two related species (D. spicata and D. palustris).
Then (P3L1-2): “As for Diodia spicata, Dessein (2003) proposed to transfer it to Galianthe based on molecular data, pollen grains (double reticulum), and fruit morphology”. Followed by “the aim of this work is to verify the position of D. palustris and D. spicata within Galianthe based on molecular and morphological data.”
The problem is that Dessein (2003) already did it. Thus (as mentioned in my previous review), I believe that the aim of the paper is rather the taxonomical transfer of D. spicata (along with its close relative D. palustris) to Galianthe. Which, by the way, is a honourable, perfectly publishable, aim.
Now, what about D. bogotensis, D. brasiliensis, D. cymosa, D. hispidula nothing is mention in the M&M or the Result sections about those four species?!?


Furthermore, before publishing this MS, I recommend the authors to go through the text again and make clearer that they consider distyly (respectively homostyly) as a morphological character state (see e.g. comments below).
In some instance homostyly or distyly is presented e.g. like a syndrome (P1L28) but without explicitly mentioning a syndrome of what? Or as “scarcely manifested” (P18L22) what does this mean?

Further comments and suggestions:

Title:
P1L2: change “Peru and the two new…” with “Peru and two new….”

Abstract:
P1L18 and L27 are in contradictions - e.g. L27 mention, “bifid stigmas and indehiscent mericaprs” has morphological characters supporting the position of Dioda palustre and D. spicata within Galianthe. But those characters are not “diagnostic features” of the genus as listed on L18-21.
P1L28. Is homostyly a syndrome? If yes a syndrome of what? Has, within this paper, homostyly is considered soly as a combination of morphological character (because there is no references to its evolution implications – in particular the transition from outcrossing to selfing. I will suggest modifying this sentence to make it clearer.

Introduction:
P2L14. “Galianthe was linked to Diodia” how was it linked and by whom? Later you say: “due to certain morphological similarities”. I would avoid this imprecise phrasing and state immediately what “link” those two genera.
P2L16-17. It is not clear to me if the authors focus on the genus Diodia or on the indehiscent mericarps?
P2L17-18. What about all those species you mentioned? Where are they now? Should they be moved to Galianthe? This is very confusing. I though that you were focusing on only two species. Why then didn’t you add those species to your phylogeny?
P2L21. “numerous” please indicate how many are “numerous”
P2L25 “several”, please be more precise and indicate how many.
P3L1. Please indicate on what bases (which morphological or molecular data) Delprete transferred those species to Spermacoce.
P3L2-4. About the aims of the paper: you state, “The aim of this work is to verify the position of D. palustris and D. spicata based on molecular and morphological data”. But this has already been done by Dessein (2003) (at least for D. spicata). Thus what are your aims? I believe (as suggested in my previous review) that your aims are to do the taxonomical transfer of D. spicata along with its close relatives to Galianthe.

Key to Galianthe species:
P16L1. I would avoid mentionning chromosome numbers in the key. This character is not observable in the field.
Furthermore, very few is known about the karyology of this group and it might well be possible that a lot of variation (e.g. aneuploidy, dysploidy, along with polyploids) occurs. Too few material have been studied in this genus to provide reliable results.

Discussion:
P18L22 what does “scarcely manifested” means. Is it distylous or not? Please explain in more details what you mean. Is that an intermediate state (a transitional state?) from distyly to homostyly (or the inverse)?
P18L13. Change: “the transference of” by the “the transfer of”

I hope those comments may be useful to the authors.

Reviewer 2 ·

Basic reporting

Authors have included most of suggestions made before. The manuscript has been improved significantly.

Experimental design

No comment.

Validity of the findings

No comment.

---

## Round 0.3 · accepted · Accept

Dear Javier,

I am glad to inform you that you study entitled "Morphological and molecular data confirm the transfer of homostylous species in the typically distylous genus Galianthe (Rubiaceae), and the description of the new species Galianthe vasquezii from Peru and Colombia" has been accepted for publication in PeerJ.
Congratulations!

Marcial.